# Impact of Next-Generation Sequencing in Diagnosis, Prognosis and Therapeutic Management of Acute Myeloid Leukemia/Myelodysplastic Neoplasms

**DOI:** 10.3390/cancers15133280

**Published:** 2023-06-22

**Authors:** Lamia Madaci, Laure Farnault, Norman Abbou, Jean Gabert, Geoffroy Venton, Régis Costello

**Affiliations:** 1TAGC, INSERM, UMR1090, Aix-Marseille University, 13005 Marseille, France; lamia.madaci@inserm.fr (L.M.); geoffroy.venton@ap-hm.fr (G.V.); 2Hematology and Cellular Therapy Department, Conception University Hospital, 13005 Marseille, France; laure.delassus@ap-hm.fr; 3Molecular Biology Laboratory, Timone University Hospital, 13005 Marseille, France; norman.abbou@ap-hm.fr (N.A.); jean.gabert@ap-hm.fr (J.G.)

**Keywords:** leukemia, myelodysplastic neoplasms, next-generation sequencing

## Abstract

**Simple Summary:**

Cytological approaches have long been used in the diagnosis, prognosis, and management of acute myeloid leukemia (AML) and myelodysplastic neoplasms. Technological advances in molecular biology, in particular next-generation sequencing (NGS), have made it possible to establish a molecular list of several gene mutations in AML and MDS, within a matter of days. The combination of cytological approaches and NGS makes it possible to pinpoint genetic mutations and cancer cell survival and proliferation pathways, enabling better clinical management and improved prognosis of patients with acute leukemia and myelodysplastic neoplasms.

**Abstract:**

For decades, the diagnosis, prognosis and thus, the treatment of acute myeloblastic leukemias and myelodysplastic neoplasms has been mainly based on morphological aspects, as evidenced by the French-American-British classification. The morphological aspects correspond quite well, in a certain number of particular cases, to particular evolutionary properties, such as acute myelomonoblastic leukemias with eosinophils or acute promyelocytic leukemias. Advances in biology, particularly “classical” cytogenetics (karyotype) and molecular cytogenetics (in situ hybridization), have made it possible to associate certain morphological features with particular molecular abnormalities, such as the pericentric inversion of chromosome 16 and translocation t(15;17) in the two preceding examples. Polymerase chain reaction techniques have made it possible to go further in these analyses by associating these karyotype abnormalities with their molecular causes, *CBFbeta* fusion with *MYH11* and *PML-RAR* fusion in the previous cases. In these two examples, the molecular abnormality allows us to better define the pathophysiology of leukemia, to adapt certain treatments (all-transretinoic acid, for example), and to follow up the residual disease of strong prognostic value beyond the simple threshold of less than 5% of marrow blasts, signaling the complete remission. However, the new sequencing techniques of the next generation open up broader perspectives by being able to analyze several dozens of molecular abnormalities, improving all levels of management, from diagnosis to prognosis and treatment, even if it means that morphological aspects are increasingly relegated to the background.

## 1. Introduction

The management of acute myeloid leukemia (AML) and myelodysplastic neoplasia (MDS) has been based, for a long time, on a cytological approach, which was already based on undiscovered molecular bases. We can mention the eosinophilia-associated features of AML4eo, for which an association with inv(16) or an equivalent was often found, without forgetting the clinical and cytological features of acute promyelocytic leukemia (APL) linked to t(15;17) or an equivalent. Among the major technological evolutions, the progress in deoxyribonucleic acid (DNA) sequencing and/or ribonucleic acid (RNA) amplification by a polymerase chain reaction (PCR) has allowed us to refine our classifications of MDS/AML in order to better understand the mechanisms of leukemogenesis, and thus, to be able to better adapt the treatments. However, until now, these tools were more or less used gene-by-gene and target-by-target. The technological evolution of next-generation sequencing (NGS) now allows us, with a delay of a few days compatible with patient management, to obtain a molecular identity card of MDS/AML involving the analysis of mutations in several dozen genes at the same time. To illustrate the interest of this new approach, the new classifications of MDS and AML, largely based on molecular data (combined with cytogenetics, classical hematological data and general evaluation of the patient’s condition, i.e., fit/unfit), should allow us, in the near future, to offer precision medicine.

## 2. Diagnosis

For the European Leukemia Net (ELN), the International Consensus Classification (ICC) and the World Health Organization (WHO) [1,2,3], molecular abnormalities take precedence over the old cytological features defined by the French-American-British (FAB) classification [4], but also over the phenotypic features defined by flow cytometry. The other clinical or biological criteria (treatment-related leukemia, pre-existing myelodysplastic or myelodysplastic/myeloproliferative neoplasms, genetic predisposition) are now only associated with the molecular diagnosis. Although the ICC and WHO classifications are in great part common, some differences can be observed (Table 1). While the ICC still uses the AML-not otherwise specified (NOS) type, the WHO delineates AML by defining genetic abnormalities vs. AML defined by differentiation. Both classifications introduce a section with other defined genetic alterations in order to allow entering novel significant molecular subtypes. Regarding the molecular subtypes, some differences exist, such as a defined section of AML with nucleoporin (NUP)98 rearrangement, RNA-binding motif protein 15 (*RBM15*), myocardin-related transcription factor-A (*MRTFA*) fusion, or other fusion genes than Myeloid/Lymphoid or mixed-lineage Leukemia, translocated to 3 (*MLLT3*) associated to lysine *N*-methyltransferase 2A (*KTM2A*) in the ICC classification, which, in contrast, is absent in the WHO classification. The most important change is that the 20% blast cutoff is not required in both classifications to define AML, except for AML with the Breakpoint Cluster Region (*BCR*): ABL Proto-Oncogene 1, Non-Receptor Tyrosine Kinase (*ABL1*) in both classifications, for CEBPA mutations in the WHO classification, AML-Myelodysplasia-related (MR) and AML-NOS defined by differentiation. Nonetheless, while the ICC has defined a cut-off at 10% for the AML with defining genetic abnormalities, the WHO has not assigned an arbitrary lower bone marrow blast cut-off if the molecular data correlates with the morphologic findings. Regarding the defining mutations of AML-MR, eight mutations are common to both classifications. The WHO does not retain the Runt-related transcription factor 1 (*RUNX1*), in contrast to the ICC. Some differences are also present regarding the AML-MR defining cytogenetic abnormalities (11q and 12p deletion in the WHO, not in the ICC, and +8 in the ICC, not retained in the WHO).

Since these molecular anomalies do not have the same prognostic values and therapeutic consequences, a hierarchy in the diagnostic classification was proposed in order to allow a clinical use of these data (Figure 1). The most obvious observation is the presence of a t(15;17) translocation, signing APL, that outweighs the other abnormalities in terms of prognosis and treatment, which is in line with the very specific therapeutic approaches (retinoic acid compounds/arsenic trioxide) used in this leukemia subtype.

In first position are the recurrent anomalies (Table 2, + other rearrangements involving the Retinoic Acid Receptor-α (*RAR α*), *KMT2A*, or *MDS1* and *EVI1* complex locus (*MECOM*)). The presence of these anomalies has another consequence, i.e., the decrease in the diagnostic threshold of AML, which is, in these cases, reduced to only ≥10% of the blasts. Regarding the cytological diagnosis, myeloblasts, monoblasts and megakaryoblasts are considered as blasts, while monoblasts and promonocytes (but not abnormal monocytes) are also classified as blasts in AML subtypes with myelomonocytic or monocytic differentiation. In the setting of APL, promyelocytes are also counted as blasts.

In the second hierarchical position are *Tumor Protein (TP)53* mutations (if variant allele frequency [VAF] ≥ 10%), which define a new type of pathology intermediate between acute leukemia and myelodysplastic syndrome (MDS/AML) if blasts are between 10 and 19%, and AML if blasts are ≥20%.

In the third position, we find (with the same distinction AML vs. MDS/AML, according to the percentage of blasts) the mutations related to myelodysplastic syndromes.

In the fourth position, we only find complex karyotypes and/or certain abnormalities related to myelodysplastic syndromes [del(5q)/t(5q)/add(5q),-7/del(7q), +8, del(12p)/t(12p)/add(12p), i(17q),-17/add(17p)/del(17p), del(20q) or idic(X)(q13)]. This downgrading shows the predominant role taken by molecular analysis (by NGS, given the number of anomalies and mutations to be detected) in the classification.

In the fifth position are the leukemias presenting none of the previous characteristics. In addition to the creation of the MDS/AML category, it is important to note the disappearance of the categories of AML with myelodysplasia-related changes (AML-MRC) and treatment-related AML. Indeed, it has been shown that it is the presence of the above-mentioned molecular abnormalities/mutations that impacts the response to treatment, as well as the prognosis, more than these clinical features. This concept is, in fact, not new since it was already known that in AML-MRC or treatment-related categories, karyotypic abnormalities were preponderant in the prognosis.

## 3. Prognosis

Regarding AML with recurrent genetic abnormalities, from a historical point of view, the most obvious prognostic value is illustrated by APL, related to *PML-RAR* translocation, which confers, even for high-risk APL, an excellent prognosis with a 2-year DFS and OS of 94% and 100% in an intention-to-treat analysis [5]. Of course, the molecular abnormality in APL has a direct effect on treatment since, in this case, more or less specific drugs are used, i.e., all-trans-retinoic acid and/or arsenic trioxide. The core binding factor AML (CBF-AML) with t(8;21) (q22;q22) or inv(16) (p13q22)/t(16;16), are considered to have a good prognosis, although the 3-year OS is 60% for inv(16) and 40% for t(8;21). These results compare favorably with other AML subtypes, however, they require novel therapeutic strategies to improve the prognosis. Some studies suggest that these CBF-AMLs benefit from fludarabine and/or gemtuzumab ozogamicin treatment in addition to the conventional 3 + 7 regimen [6]. In AML with 12q23.3/*KMT2A* rearrangements, some additional gene mutations can be detected by NGS involving the RAS pathway in one-third of patients. The analysis of the type of rearrangement shows that patients with AML t(9;11)(p21.3;q23.3)/*MLLT3::KMT2A* have a better prognosis than those with other 11q23/*KTM2A* rearrangements [7]. In AML with t(6;9)(p22.3;q34.1)/*DEK::NUP214*, the translocation can involve the two proto-oncogenes *SET* and *DEK*, resulting in a *SET-NUP214* and *DEK-NUP214* chimeras, whose pathophysiological mechanism relies on the properties of the resulting fusion protein. These fusion proteins disrupt nuclear protein export by the inhibition of Chromosomal Maintenance 1/Exportin 1 (CRM1). Nonetheless, CRM1 drug-targeted inhibition relocalizes SET-NUP214 and DEK-NUP214 nuclear bodies throughout the nucleoplasm and exerts an anti-leukemic effect in AML with NUP214 rearrangement [8]. This suggests that the use of selective inhibitors of nuclear export (SINEs) could be of interest in these AMLs and are tested in clinical trials. AMLs with inv(3)(q21.3q26.2) or t(3;3)(q21.3;q26.2) have the classic ribophorin(*RPN*)*1-MECOM* rearrangement with *MECOM(EVI1)* over-expression and *GATA2* haploinsufficiency. The 3q-AML are very sensitive to the inhibition of Poly [ADP-ribose] polymerase 1 (PARP1) that decreases MECOM(EVI1) expression, providing an interesting strategy for these AMLs [9]. The *nucleophosmin (NMP)-1* mutated AMLs are of great interest since they are the most frequent (one-third) mutation in AML, even more frequent than AML with a normal karyotype (>50%). The *NPM1* is a nucleus-cytoplasmic shuttling protein. This AML subtype is considered to have a good prognosis. Attempts to further improve the prognosis involve the use of bcl-2 antagonists (venetoclax) or more specific approaches aiming at modifying the structure of *NPM1*, the level of the protein or its effects in the nucleolus, more particularly via Menin inhibitors [10,11]. The requirement for a very precise molecular AML characterization is illustrated by the AML with mutated CCAAT/enhancer-binding protein alpha (*CEBPA*). Interestingly, in-frame basic leucine Zipper (bZIP) mutated CEBPA confers a good prognosis, while mutations of *CEBPA* out of the bZIP domain are not associated with a good prognosis [12]. Nonetheless, the good prognosis of AML with recurrent genetic abnormalities CBF-AML is modified by other mutations, such as *Fms-Like Tyrosine kinase 3 (FLT3)-Internal tandem Duplication* (*ITD*), demonstrating, again, the usefulness of NGS analysis [13]. The interest of molecular biology in both diagnosis, prognosis and treatment in AML with t(9;22)(q34.1;q11.2)/*BCR::ABL1* is obvious since specific targeted therapies are already available and used both in chronic myeloid leukemia (CML) and AML, with a drastic improvement of prognosis, more particularly for patients unfit for allogeneic stem cell transplantation.

Recently, a novel entity has been defined among AML, i.e., AML with mutated *TP53*, with or without the loss of wild-type *TP53* detected by cytogenetic FISH. In these AMLs, the *TP53* mutation overcomes the other genetic abnormalities, which are very frequent in this AML subtype and often associated with a complex karyotype. The NGS approach allows an in-depth clinical and molecular analysis of *TP53* mutations, as demonstrated by the study of Grob et al. [14], which included a huge number (2200) of AML/MDS with an excess of blast (AML/MDS-EB). More than 10% of AML/MDS-EB had mutated *TP53* with a bi-allelic mutation in 76% of these cases, with an associated mutation in half of the patients. Of note, no difference in the *TP53* status was observed between AML and MDS-EB patients, and in parallel, no differences in survival were detected between AML and MDS-EB. Although NGS sequencing allowed to describe and clearly define the novel entity TP53 AML/AMDS-EB, the results of *TP53* measurable residual disease (MRD) monitoring are disappointing since no correlation was observed with survival, despite using deep sequencing that detected MRD positivity in 73% of patients in complete remission (CR). This suggests the persistence of residual *TP53* mutated clones not detected by bulk sequencing, requiring a single-cell approach [15] or more sensitive techniques that are discussed in the MRD section. The new classifications related to *TP53* pose a dilemma in terms of treatment [16]. The hypomethylating agents are a standard approach to MDS, however, they confer less than 20% of CR, with an overall survival of less than one year. When patients have deleted *TP53*, or mutated *TP53* with >40% VAF, the prognosis is even worse, with an overall survival of about 6 months, partly due to resistance to bcl2 inhibition. Due to the modifications in AML classification, the mutation of TP53 defines a novel entity that can benefit from specific therapeutic approaches. Among the most interesting drugs, decitabine seems to be of great interest (in association with venetoclax, 48% CR) despite a severe toxicity when using a ten-day regimen [17]. An impressive CR of 64% is obtained with the association of magrolimab (an anti-CD47 monoclonal antibody macrophage checkpoint inhibitor) with venetoclax and azacitidine [16]. The association of sabatolimab, which targets T-cell immunoglobulin and mucin-domain containing-3 (TIM-3), with hypomethylating agents confers a duration of response (DOR) that approaches 2 years [16]. Other investigational approaches are under consideration, including Chimeric Antigen Receptor (CAR)-T-cell, antibodies against various myeloid antigens, and immune checkpoint inhibitors or drug-targeting mutant TP53, such **as nanomolar-affinity pharmacological chaperones that stabilize the oncogenic mutant molecule** [18].

The third hierarchical level of the ICC classification also relies on mutation detection via NGS. The retained significant mutations are *ASXL1*, *BCOR*, *EZH2*, *RUNX1*, *SF3B1*, *SRFS2*, *STAG2*, *U2AF1* and/or *ZRSR2* (Table 3). Of note, these mutations define AML with MDS-related gene mutations even in the absence of a previous history of cytological signs of MDS, pointing again to the pivotal role of NGS sequencing that overcomes both clinical history and cytological considerations.

■*ASXL1* is part of Enhancers of Trithorax and Polycomb (ETP) proteins that assemble chromatin modification complexes and transcription factors. In contrast to *ASXL2* mutations [often associated with t(8;21)], *ASXL1* confers a poor prognosis. Of note, this mutation can orientate the therapeutic intervention since some experiments suggest that *ASXL1* mutants confer sensitivity to BET bromodomain inhibitors (BETis) and resistance to Histone DeACetylase inhibitors (HDACi) [19].■The *BCOR* gene encodes for a subunit of the Polycomb Repressive Complex (PRC) 1.1, one of the six complexes that constitute the PRC1. By pairing the analysis of 6162 patient samples with cell line experiments, Schaefer et al. [20] have shown that the *BCOR* mutation inactivates the repressive activity of PCR1.1, leading to aberrant oncogenic signaling, suggesting that specific inhibitors of the RAS/MAPK pathway could be of interest in this AML subset.■*The Enhancer of Zeste Homolog 2* (*EZH2*) mutation corresponds to the loss of function of a histone methyltransferase activity that participates in the Polycomb Repressive Complex 2 (PRC2), whose role is to silence genes involved in stem cell renewal, including leukemic stem cells (LSC). Interestingly, the identification of *EZH2* mutations was correlated, in most cases, with the loss of EZH2 protein expression, as evaluated by immunohistochemical analysis, thus directly linking the sequencing data to the AML phenotype [21].■The *RUNX1* gene, as previously seen, is involved/translocated in some recurrent genetic abnormalities in AMLs (of good prognosis), although it can also be mutated in 10% of MDS or AMLs. In this case, a loss of function is the common consequence of the mutation (missense/deletion truncation), and mtRUNX1 confers a poor prognosis. The mutation of *RUNX1* impairs ribosome biogenesis (RiBi), leading to the hypothesis that drugs that perturb RiBi could be of interest in that AML subset. In accordance, AMLs with mtRUNX1 show an important sensitivity to homoharringtonine and BET inhibitors, with a synergistic effect of the B-cell lymphoma 2 (BCL2) inhibitor venetoclax, suggesting a specific chemotherapy regimen for these AMLs [22].■The *Splicing Factor 3B subunit 1 (SF3B1)* mutation corresponds to most “sideroblastic anemias” of the previous classification and is considered MDS-related. This mutation is not restricted to the myeloid lineage since it is found in CLL, T–ALL and solid tumors, such as melanoma, pancreatic adenocarcinoma or breast cancer. The main function of SF3B1 is the assembly of the spliceosome, so many inhibitors of the spliceosome are under investigation in AML or MDS [23].■The *Serine and Arginine Rich Splicing Factor 2 (SRSF2)* is a splicing factor in which mutations occur in up to 20% of AML/MDS and is frequently associated with mtRUNX1, conferring a particularly poor prognosis [24]. The reasons why the mtSRSF2/mtRUNX1 association is so deleterious have been analyzed [25]. Very interestingly, this co-mutation induces mis-splicing of a series of genes involved in the DNA damage response and in the cell-cycle checkpoint pathways (Fanconi anemia of complementation group J gene/*BRIP1*, *NABP1*, *TBRG4* and *AKAP8L*). This example demonstrates the usefulness of a comprehensive and simultaneous analysis of all these mutations in order to take account of their possible cooperative effects on AML pathophysiology.■*STromal AntiGen 2 (STAG2)* is one of the core components of the cohesin complex, a global protective structure around DNA molecules, whose mutation usually leads to a loss of function. The mutation of *STAG2* induces a switch of the cohesion complex to use *STAG1*, which is rarely mutated in AML/MDS [26]. This switch induces global alterations of chromatin, and from a therapeutic point of view, an interesting consequence, i.e., a 70-fold increase in sensitivity to poly(ADP-ribose) polymerase (PARP) inhibitors, with some compounds being under investigation, such as talazoparib. Of note, some data also predict a synergy with hypomethylating drugs, suggesting a novel “backbone” associated (cf. previous mutations) with various partners depending on the other mutations identified by NGS analysis.■The *U2 small RNA nuclear auxiliary factor 1 (U2AF1)* plays a central role in the alternative splicing of pre-mRNA and is mutated in 10% of AML/MDS. It is considered a poor prognosis marker, and although no specific drugs target this molecule, splicing modulators may be of interest in this AML subtype [27].■The *ZRSR2 (Zinc Finger CCCH-Type, RNA Binding Motif and Serine/Arginine Rich 2)* is a splicing factor rarely mutated (5% of MDS/AML) and often associated with mutation of the epigenetic regulator TET2. Its various mutations lead to a loss of function. Interestingly, although many abnormalities are detected in ribosome functions, inflammation or cell mobility, a major dysregulation of the MAPK pathway seems to play a central role in leukemogenesis [28], suggesting the use of inhibitors [29].

The fourth hierarchical level of ICC classification corresponds to MDS/AMLs with a complex karyotype, with a more or less classical poor prognosis abnormality, such as del(7q) or −7. Of note, before the NGS era, cytogenetic data were clearly positioned at the first hierarchical level of AML classification, further underlining the current central role of NGS in MDS/AML management.

Finally, few MDS/AMLs will correspond to the fifth hierarchical level of MDA/AML, not otherwise specified in the exhaustive molecular and cytogenetic analysis, which are mandatory for myeloid malignancies management.

## 4. Impact of the Novel Classification on Myelodysplastic Syndrome: The M-IPSS

Among the most interesting impacts of the availability of NGS data is the possibility to refine the prognosis for each patient while evaluating both the classical hematological data (cytopenia, blast counts, age), cytogenetic data and the large mutational analysis data. This led to the elaboration of the Molecular International Prognostic Scoring System (M-IPSS) [30]. The clinical-molecular prognostic model M-IPSS was developed from 2957 MDS patients (604 patients were included in the data set) with a mutation screening of 152 genes. The M-IPSS was secondly tested on a validation cohort of 754 patients. Finally, the score was calculated using hematological parameters, cytogenetic and somatic mutations of 31 genes, leading to six different prognosis categories. In comparison with the previous R-IPSS, 46% of patients were stratified, of whom 74% were upstaged and 26% down-staged. The median OS ranged from 10.6 years (very low M-IPSS) to 1.0 (very high M-IPSS), with death by 4 years, from AML-t, ranging for the same categories from 2.8% to 42.8%, and death without AML from 15.9% to 51.3%. This shows that the M-IPSS not only predicts death by AML but includes other causes, nonetheless, with a less discriminant potency. The *TP53*multihit mutations, *KMT2A (MLL)* partial tandem duplication (PTD), and *FLT3-ITD* correlated with the worse prognosis, while *ASXL1*, *BCOR*, *EZH2*, *NRAS*, *RUNX1*, *STAG2* and *U2AF1* mutations were also associated with adverse risks. The good prognosis of SF3B1 was modified by the presence of co-mutations. Clearly, the analysis of prognosis is not directly feasible by the clinician since it requires taking into account all the mutations and their co-occurrence. Consequently, a useful M-IPSS Web calculator, easily available, has been developed (https://mds-risk-model.com, accessed on 27 October 2022) that also includes missing values and is both applicable for primary and secondary/therapy-related MDS.

## 5. Measurable Residual Disease (MRD)

The evaluation of MRD can be done with various techniques, mainly using multiparameter flow cytometry (MPFC), real-time polymerase chain reaction (RT-PCR) or NGS [31]. The impact of NGS has to be carefully evaluated. The study by McGowan et al. [31] involved 107 patients and compared 717 MPFC tests to 247 NGS tests. The evaluation of MRD by MPFC was based on the detection of the initial aberrant phenotype by analyzing 500,000 events, while NGS MRD used a panel targeting 141 myeloid-related genes. The patient’s mutations corresponded mainly to *FLT3*/*NPM1*, *KIT*/*NRAS*/*KRAS*/*PTPN11, GZTA2*/*CEBPA*/*WT*, *TP53*/*IDH1*/*IDH2*/*RUNX1*, or *DNMT3A*/*ASXl1*/*TET*, among others. Regarding the 247 instances with both MPFC and NGS MRD tests, there were 197 MPFC+/NGS+ instances, 3 MPFC−/NGS− instances, 44 MPFC−/NGS+ instances and 3 MPFC+/NGS− instances; thus, there was a 19% (47 out of 247) discrepancy between the two methods. Of note, these discrepant results were mainly observed with mutations not associated with adverse outcomes (*DNMT3A*/*ASXL1*/*TET* or others, 17 tests), mutations of unknown significance (14 tests), mutations that were likely pathogenic (3 tests) and 1 patient without mutations. If we exclude these tests, regarding the remaining 131 pairs, 12 are discrepant with 2 MPFC+/NGS− and 10 MPFC−/NGS+. Although the study sample size is limited, these data suggest that NGS may avoid missing low-volume MRD. Nonetheless, the clinical impact of such findings is to be evaluated in controlled trials so that NGS cannot directly impact the treatment that still relies on MPFC and RT-qPCR MRD evaluation. Another impact of NGS is the level of sensitivity of the method used for MRD assessment. The predictive value of NGS vs. MPFC MRD was analyzed and compared in the study by Tsai et al. [32]. This study tested MRD via two different methods in 335 patients with de novo AML in morphological remission (excluding APL) after the first chemotherapy course and after the first consolidation course. The first conclusion was that the study of the common Clonal Hematopoiesis of Indeterminate Potential (CHIP)-related gene mutations (*DNMT3A*, *TET2* and *ASXL1*) was not informative for MRD assessment. Then, this study showed that the MRD assessment at time 1 by NGS was not predictive of relapse if MPFC MRD was negative but had a poor prognosis value at time 2 even if MPFC was still negative. Due to the rise of specifically targeted therapy, the specific assessment of druggable mutations is of pivotal interest. One of the most common pathogenic mutations detected in AML is *FLT3-ITD* (25% of patients), with various targeted drugs, such as midostaurin or gilteritinib. The detection of *FLT3-ITD* MRD is challenging and is mainly done via fragment analysis (FA) with a detection sensitivity of 2%. Recently, the development of novel NGS analysis methods has resulted in a 0.001% sensitivity that was superior to FA analysis [33]. The clinical impact of this higher sensitivity was questioned in the study by Loo et al. [34]. The MRD positivity before the allogeneic transplant was 37% when evaluated by NGS vs. 6.7% by the conventional technique. Of note, the MRD detected by NGS, even at a very low level, had a dismal prognosis, with a 2-year RFS of 78%, 32%, 40% and 0% for NGS MRD level < 0.001 (negative), 0.001–0.1%, 0.1–1% and >1% respectively. In conclusion, *FLT3-ITD* MRD detection by NGS is an important prognosis factor that can precisely define the relapse risk.

## 6. Discussion

Since the screening for an entire set of genes is now mandatory for the recent MDS/AML ELN/ICC/WHO classifications [1,2,3], that locates NGS technology at a pivotal position for the management of MDA/AML. It is clear that we move from the chemotherapy area, which kills all the dividing cells, towards molecular medicine, which targets gene mutations shown by the NGS approach. For the clinician, the analysis of such numerous data cannot be done intuitively and requires the use of dedicated calculators that can synthesize classical data, such as bone marrow blasts, cytopenias, cytogenetic and molecular data in order to define the patient’s prognosis to choose the best therapeutic option (https://mds-risk-model.com, accessed on 27 October 2022). Some pivotal advances in MDS/AML management have been voluntarily not discussed in this article, more particularly concerning druggable mutations, such as *FLT3*, *IDH1* or *IDH2*, since they have been extensively reviewed. Of note, data from NGS are useful for diagnosis in order to select the best therapeutic options for patients (for review, [1,35,36,37]). Nonetheless, MDS/AML tumoral cell analysis is only one part of the problem, with the other part being the patient itself and his possibility to support intensive treatment. As a consequence, the classification of the patient in three simple categories (go-go, slow-go or no-go) is of utmost importance in order to propose the most adequate treatment [38].

Finally, it is useful to have a historical view of NGS in AML via some milestone publications. In 2012, Patel et al. [39] published a seminal paper analyzing 18 genes in 398 patients <60 years that randomly received high-dose or standard-dose daunorubicin. Genetic predictors defined patients who should benefit from high-dose daunorubicin (*DNMT3A* or *NPM1* mutations or *MLL* translocations). In 2016, a different approach was used, relying not on mutation detection as in the previous paper but on the identification of a 17-gene stemness transcriptional signature that predicted a poor prognosis even when allogeneic transplantation was performed [40]. Nonetheless, in order to refine the prognosis factor, more and more genes enter the predictive models and require machine learning to be usefully analyzed. The analysis of the Oregon Health & Science University (OHSU) and of The Cancer Genome Atlas (TCGA) datasets allowed us to identify 197 common “protective” genes and 87 common “risk” genes between the two databases with a very potent predictive value [41].

## 7. Conclusions

The next challenge now will be in the elaboration of predictive models to be able to account for all the molecular markers, i.e., fusion gene transcripts, mutations, and gene expression. Nonetheless, all these efforts have a sense only if they are able to develop precision medicine, allowing the introduction of therapy that reaches a good target. More precisely, targeting a single mutation (as tyrosine kinase inhibitors for *BCR::ABL1* fusion gene, *FLT3* or *IDH1/2* mutations) is not the “magic bullet” since leukemia heterogeneity will allow tumor escape via clonal drift under chemotherapy pressure. Probably the next step of the NGS approach in AML should require single-cell RNA or DNA analysis, thus requiring more and more bioinformatics resources to allow their routine clinical use. Advances in sequencing speed and steadily decreasing costs suggest that this approach could become part of routine AML assessment in the future. In combination with this highly targeted approach, the use of drugs aimed at targeting pathways central to the survival of leukemia cells, and leukemic stem cells in particular [42], will help improve the still rather poor prognosis of acute leukemia.

## Figures and Tables

**Figure 1 cancers-15-03280-f001:**
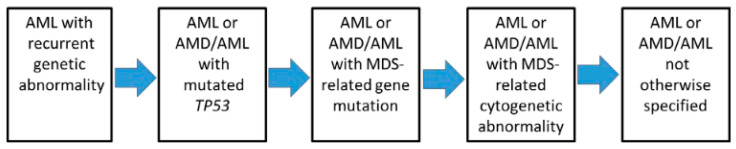
Hierarchical classification of the International Consensus Classification of AML (from [1]).

**Table 1 cancers-15-03280-t001:** Comparison of WHO and ICC classifications for AML.

WHO 2022	ICI
Acute myeloid leukemia with defining genetic abnormalities Acute promyelocytic leukemia with PML::RARA fusion Acute myeloid leukemia with RUNX1::RUNX1T1 fusion Acute myeloid leukemia with CBFB::MYH11 fusion Acute myeloid leukemia with DEK::NUP214 fusion Acute myeloid leukemia with RBM15::MRTFA fusion Acute myeloid leukemia with BCR::ABL1 fusion Acute myeloid leukemia with KMT2A rearrangement Acute myeloid leukemia with MECOM rearrangement Acute myeloid leukemia with NUP98 rearrangement Acute myeloid leukemia with NPM1 rearrangement Acute myeloid leukemia with CEBPA rearrangement Acute myeloid leukemia, myelodysplasia-related Acute myeloid leukemia with other defined genetic alterations	Acute myeloid leukemia with defining genetic abnormalities Acute promyelocytic leukemia with PML::RARA fusion or other RARA rearrangement Acute myeloid leukemia with RUNX1::RUNX1T1 fusion Acute myeloid leukemia with CBFB::MYH11 fusion Acute myeloid leukemia with DEK::NUP214 fusion XXX Blast phase CML Acute myeloid leukemia with KMT2A rearrangement: MLLT3 Acute myeloid leukemia with MECOM rearrangement XXX Acute myeloid leukemia with NPM1 rearrangement Acute myeloid leukemia with CEBPA rearrangement Acute myeloid leukemia, myelodysplasia-related Acute myeloid leukemia with other defined genetic alterations
Acute myeloid leukemia, defined by differentiation Acute myeloid leukemia with minimal differentiation Acute myeloid leukemia without maturation Acute myeloid leukemia with maturation Acute basophilic leukemia Acute myelomonocytic leukemia Acute monocytic leukemia Acute erythroid leukemia Acute megakaryoblastic leukemia	AML, not otherwise specified XXX XXX XXX XXX XXX XXX XXX
MDS-MR defining mutations ASXL1 BCOR EZH2 RUNX1 SF3B1 SRSF2 STAG2 U2AF1 ZRSR2	MDS-MR defining mutations ASXL1 BCOR EZH2 XXX SF3B1 SRSF2 STAG2 U2AF1 ZRSR2

**Table 2 cancers-15-03280-t002:** Recurrent genetic abnormalities (ELN classification).

APL with t(15;17)(q24.1;q21.2)/PML::RARA
AML with t(8;21)(q22;q22.1)/RUNX1::RUNX1T1
AML with inv(16)(p13.1q22) or t(16;16)(p13.1;q22)/CBFB::MYH11
AML with t(9;11)(p21.3;q23.3)/MLLT3::KMT2A
AML with t(6;9)(p22.3;q34.1)/DEK::NUP214
AML with inv(3)(q21.3q26.2) or t(3;3)(q21.3;q26.2)/GATA2, MECOM(EVI1)
AML with other rare recurring translocations
AML with mutated NPM1
AML with in-frame bZIP mutated CEBPA
AML with t(9;22)(q34.1;q11.2)/BCR::ABL1

**Table 3 cancers-15-03280-t003:** Myelodysplasia-related gene mutations (ELN classification).

ASXL1	Additional Sex Combs-Like 1, Transcriptional Regulator
BCOR	BCL6 CoRepressor
EZH2	Enhancer of zeste homolog 2, H3 histone methylation
RUNX1	Runt-related transcription factor 1, core binding factor family
SF3B1	Splicing Factor 3B subunit 1
SRSF2	Serine and arginine Rich Splicing Factor 2
STAG2	Stroma AntiGen 2, cohesin complex
U2AF1	U2 small nuclear RNA Auxiliary Factor 1, splicing factor
ZRSR2	Zing finger CCCH-type, RNA binding motif and Serine/arginine Rich 2, splicing factor

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
