# Peer review of "Impact of Next-Generation Sequencing in Diagnosis, Prognosis and Therapeutic Management of Acute Myeloid Leukemia/Myelodysplastic Neoplasms"

_cancers, 2023, doi:10.3390/cancers15133280_

Round 1

Reviewer 1 Report

The study demonstrates the possibility of the next generation techniques in the application of diagnosis of acute myeloid leukemia/myelodysplastic neoplasms.

Table 2 and 3 needs to be revised to correctly show the reccurente anomalies and mutations related to myelodysplastic syndromes.

Author Response

Reviewer N°1

-   Table 2 has been checked, it corresponds to a part of the Table 1 in the ELN classification. Nonetheless, rare abnormalities (added in foot notes in the ELN article) have not been reported in our review, to simplify reading.

-    Table 3 title has been changed to « Myelodysplasia-related gene mutations », that more precisely corresponds to the cited mutations.

Reviewer 2 Report

In this review, the authors supported that impact of next generation sequencing in diagnosis, prognosis and therapeutic management of acute myeloid leukemia/myelodysplastic neoplasms. This is an interesting topic with significant prospect of clinical application. The new sequencing techniques of the next generation (NGS) open up broader perspectives by being able to analyze several dozen of molecular anomalies.This review is well performed study in the important role of NGS in AML/MDN. However, the manuscript requires some revisions.

1. In line 202-203, the retained significant mutations are ASXl1, BCOR, EZH2, RUNX1, 202  (Table 2) does not correspond to the table 2 (In line 88), please confirm that.

2. Please define the abbreviations in the text when used for the first time. For example, in line 34 AML” .

Author Response

Reviewer N°2

 - In line 202-203, “the retained significant mutations are ASXl1, BCOR, EZH2, RUNX1… » does not correspond to Table 2 but to Table 3. This error has been corrected. I apologize for this mistake.

- We checked the abbreviations in order to define them in the text when used for the first time.

Reviewer 3 Report

Lamia Madaci and co-authors present a high quality and well-written review manuscript focused on the impact of next generation sequencing in diagnosis, prognosis and therapeutic management of acute myeloid leukemia/myelodysplastic neoplasms.

Authors suggest that advances in biology, particularly "classical" cytogenetics (karyotype) and molecular cytogenetics (in situ hybridization) have made it possible to associate certain morphological features with particular molecular abnormalities, such as pericentric inversion of chromosome 16 and translocation t(15;17) in the two preceding examples. PCR techniques have made it possible to go further in these analyses, by associating these karyotype abnormalities with their molecular causes, CBFbeta fusion with MYH11 and PML-RAR fusion in the previous cases. In these two examples, the molecular abnormality allows to better define the pathophysiology of the leukemia, to adapt certain treatments (alltransretinoic acid for example), and to follow up the residual disease, of strong prognostic value beyond the simple threshold of less than 5% of marrow blasts signaling the complete remission. However, the new sequencing techniques of the next generation open up broader perspectives by being able to analyze several dozen of molecular anomalies, improving all levels of management, from diagnosis to prognosis and treatment, even if it means that morphological aspects are increasingly relegated to the background. 

Authors cover such aspects as: diagnosis, prognosis, impact of the novel classification on myelodysplastic syndrome, measurable residual disease. 

Finally, authors conclude that advances in sequencing speed and steadily decreasing costs suggest that this approach could become part of routine AML assessment in the future. In combination with this highly targeted approach, the use of drugs aimed at targeting pathways central to the survival of leukemia cells, and leukemic stem cells in particular, will help improve the still rather poor prognosis of acute leukemia. 

Overall, the manuscript is highly valuable for the scientific community and should be accepted for publication.

======================

Other comments to authors:

1) Please check for typos throughout the manuscript.

2) With regards to TP53 mutations – authors are kindly encouraged to cite the following article that describes the development of novel therapeutics targeting mutant p53. DOI: 10.1021/acsptsci.2c00164

Author Response

Reviewer N°3

- We have checked for typos throughout the manuscript.

- As suggested, the following reference has been added to our article (ref 18) ; Discovery of Nanomolar-Affinity Pharmacological Chaperones Stabilizing the Oncogenic p53 Mutant Y220C , J. R. Stephenson Clarke, L. R. Douglas, P. J. Duriez, D. I. Balourdas, A. C. Joerger, R. Khadiullina, et al. ACS Pharmacol Transl Sci 2022 Vol. 5 Issue 11 Pages 1169-1180

 - The following sentence has bee, added ; « nanomolar-affinity pharmacological chaperones that stabilize the oncogenic p53 mutant. »

Round 2

Reviewer 1 Report

The description of TP53 mutations may be revised to clarify "blasts" more in detail in 2. Diagnosis in the second paragraph of page 4

The description of positions may be proofread in the paragraphs of page 2. Careful proofreading is needed.

Author Response

1) the definition of blasts, as detailed in figure 1 of the ELN article, has been added in the paragraph:

Regarding the cytological diagnosis, myeloblasts, monoblasts and megakaryoblasts are considered as blasts, while monoblasts and promonocytes (but not abnormal monocytes) are also classified as blasts in AML subtypes with myelomonocytic or monocytic differentiation. In the setting of APL, promyelocytes are also counted as blasts. 

2) the description of positions has been modified in order to fit to Cancers publication use: 

1TAGC, INSERM, UMR1090, Aix-Marseille University, 13005 Marseille, France.

2Hematology and Cellular Therapy Department, Conception University Hospital, 13005 Marseille, France.

3Molecular Biology Laboratory, Timone University Hospital, 13005 Marseille, France.